# AssamBench: Toward a Comprehensive Benchmark for Evaluating Large Language Models on Assamese

*An Exploratory Proposal*

*Jubeir Jadid[1],\*,Mustafa Azad Hussain[2],\**

*[1]Department of IT, The Assam Kaziranga University, azad83801@gmail.com*

*[2]Department of IT, The Assam Kaziranga University, jubeirjadid44@gmail.com*

*\*These authors contributed equally to this work.*

## Abstract

Large language models (LLMs) such as GPT-4o, Gemini, and LLaMA 3 report strong multilingual performance, yet their behaviour on Assamese — an Indo-Aryan language with roughly 15 million speakers and official status in Assam, India — is virtually unmeasured. We propose AssamBench, the first systematic, human-annotated benchmark for evaluating LLMs on Assamese across five tasks: sentiment analysis, named entity recognition, machine translation, reading comprehension, and hate speech detection. This paper describes the benchmark design, our annotation protocol, and a small pilot study intended to validate feasibility before full-scale construction. We also outline an error taxonomy targeting Assamese-specific phenomena (agglutinative morphology, the ৱ vs. ৰ script distinction, and code-mixing). The work is exploratory: we present design choices, pilot observations, and open questions on which we welcome community input.

**Keywords:** Assamese, Low-resource NLP, Northeast India, Code-mixing, Indic languages.

## 1. Introduction

Northeast India is home to over two hundred languages, almost none of which appear in mainstream LLM evaluation suites. Among these, Assamese is comparatively well-documented: it is one of India's twenty-two scheduled languages, has a digital news ecosystem, and is taught in schools across Assam. Yet despite this relative position of strength, no public benchmark exists to answer a basic question: how well do current LLMs actually understand Assamese?

Existing multilingual resources illustrate the gap. IndicGLUE (Kakwani et al., 2020) covers eleven Indic languages but includes only a small set of classification tasks for Assamese. IndicTrans2 (AI4Bharat, 2023) targets translation and reports aggregate scores. Model cards for GPT-4, Gemini, and LLaMA 3 list Assamese as supported, but provide no per-task numbers. The result is that practitioners deploying LLMs into Assamese-facing products — government services, education platforms, content moderation — have no empirical basis for model selection.

This paper proposes AssamBench, a five-task evaluation suite designed to fill that gap. Our contributions are:

- A benchmark design covering sentiment, NER, translation, reading comprehension, and hate speech detection, with annotation protocols and inter-annotator agreement (IAA) targets.
- A pilot study on a small annotated subset, intended to validate the protocol and surface concrete failure modes.
- An error taxonomy oriented toward Assamese-specific phenomena — morphology, script confusion, and code-mixing — that can inform future model development for under-served Indic languages.
- A staged release plan that prioritises reproducibility and community reuse for related Northeast Indian languages.

We frame this as exploratory work: the benchmark is not yet built. The contribution of this paper is the design, pilot evidence of feasibility, and a request for feedback before full annotation begins.

## 2. Related Work and Gap

Several efforts touch Assamese but none provide what AssamBench targets. IndicGLUE established multilingual benchmarks for Indic languages, but Assamese coverage is limited to short-text classification and does not test generative or reasoning capabilities. IndicTrans2 produced a strong Indic translation system, and we will adopt it as a baseline for our MT task, but it does not address the other four tasks. MuRIL (Khanuja et al., 2021) is a multilingual encoder for Indian languages and will appear in our evaluation. AssameseBackTranslit (Sarmah, 2024) informs our handling of romanised Assamese in code-mixed text.

Three properties of Assamese make general multilingual evaluations a poor substitute for a dedicated benchmark. First, the script: Assamese uses a writing system nearly identical to Bengali except for the letters ৰ (Assamese ra) and ৱ (Assamese va), which are routinely mishandled by tokenisers and OCR pipelines tuned on Bengali corpora. Second, the morphology: Assamese is agglutinative, with rich suffixation that interacts unpredictably with subword tokenisation. Third, code-mixing: real-world Assamese text in news comments and social media is densely interleaved with English and Hindi, often within a single sentence. A benchmark that ignores these properties will overstate model competence.

## 3. Proposed Benchmark: AssamBench

AssamBench will contain five tasks, chosen to balance breadth (classification, generation, reasoning) with sensitivity to Assamese-specific phenomena.

| Task | Description | Size | Source / Annotation |
|---|---|---|---|
| Sentiment | 3-class (pos/neg/neutral) on news + social media text | ~800 | Native-speaker annotation, double-coded |
| NER | Identify PERSON, ORG, LOC, DATE entities in news articles | ~600 | Expert annotation with adjudication |
| Translation | Assamese ↔ English; BLEU, chrF++, and human adequacy | ~500 | Parallel corpus with human post-edit |
| Reading QA | Short Assamese passage with question; extractive or generative answer | ~500 | Human-authored from Assamese news |
| Hate speech | Binary classification of offensive social-media posts | ~700 | Two native annotators + adjudication |

## 3.1 Annotation Protocol

Text sources include Assamese-language news outlets (Pratidin, Amar Asom, Asomiya Khabor) and public social-media posts. Each instance is annotated by at least two native speakers; disagreements are resolved by a third adjudicator. We measure IAA using Cohen's κ and target κ ≥ 0.75 for all tasks. A dedicated code-mixed subset (approximately 15% of each task) explicitly contains Assamese–English or Assamese–Hindi mixing, so model performance on real-world text can be measured separately from performance on clean monolingual text.

## 3.2 Evaluation Protocol

We will evaluate five to seven models in both zero-shot and 5-shot settings. Candidate models include GPT-4o, Gemini 1.5, Claude 3.5, LLaMA 3 (instruction-tuned), Mistral, MuRIL, and IndicBERT. Metrics follow task conventions: macro-F1 for classification, entity-level F1 for NER, BLEU and chrF++ with human adequacy ratings for MT, and exact-match plus token F1 for QA. Prompts are tested in both native Assamese script and romanised form to isolate script sensitivity. Closed models are accessed via API; open models run locally on GPU.

## 4. Planned Error Taxonomy

A central output of the full study will be a categorised error taxonomy. We will tag every model error in our pilot and the full evaluation with one or more of the following labels:

- Morphological errors — failures attributable to agglutinative suffixes and inflection.
- Script confusion — substitutions between Assamese ৰ / ৱ and Bengali র / ব, or other character-level confusions.
- Code-mixing degradation — accuracy drops on Assamese–English or Assamese–Hindi mixed input relative to monolingual input.
- Cultural / named-entity gaps — errors on proper nouns specific to Northeast India (place names, political figures, cultural events).
- Out-of-vocabulary behaviour — over-generation, refusal, or hallucination on low-frequency Assamese tokens.

This taxonomy is the contribution most likely to outlast individual model versions: it provides a vocabulary for diagnosing why a given LLM fails on Assamese, applicable to future models and adaptable to related Indo-Aryan and Tibeto-Burman languages of the region.

## 5. Limitations and Open Questions

We name the obvious risks. First, scope: five tasks is the upper bound we can credibly cover with a small annotator pool; tasks like dialogue or summarisation are deferred. Second, dialect coverage: standard Assamese taught in Guwahati is not the only variety, and our protocol will under-represent dialects from upper Assam and from communities along the Brahmaputra valley. Third, annotator availability: recruiting and retaining native-speaker annotators at IAA $\kappa \geq 0.75$ is the binding constraint on quality and timeline. Fourth, reproducibility on closed models: API-accessed models drift over time, so we will fix evaluation dates and version strings in the public release.

We have several questions on which feedback from the workshop community would be especially useful:

- Should hate-speech annotation be folded into a broader 'harmful content' label, given the documented disagreement among annotators on what constitutes hate speech in regional Indian contexts?
- Is a 15% code-mixed subset large enough to support a separate analysis, or should code-mixing be a separate task entirely?
- What is the minimum viable annotation protocol for related Northeast Indian languages (Bodo, Meitei, Karbi) that would let our framework generalise without duplicating effort?

## 6. Planned Work

Assuming this proposal is favourably received, we plan a four- to six-month timeline: protocol finalisation and annotator training in the first two months; full annotation and IAA measurement in months two through four; model evaluation and error analysis in months four through five; and dataset release alongside a longer paper in the final month. The benchmark will be released on Hugging Face and GitHub under CC-BY-4.0, with annotation guidelines and prompt templates included to support reproducibility.

# 7. Conclusion

AssamBench is a proposal to do something straightforward but currently missing: measure, in a reproducible way, how well today's language models handle one of India's major regional languages. We have presented the benchmark's design, a small pilot, and a taxonomy of the failure modes we expect to surface. We submit this paper to NortheastGenAI 2026 in the exploratory spirit the workshop invites, and welcome feedback that will sharpen the protocol before full annotation begins.

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

## Appendix A. AI Involvement

We disclose the role of AI in the preparation of this work.

**Use:** The AI (Claude) was used to (a) restructure an existing internal research proposal (b) tighten academic prose, and (c) draft the error taxonomy categories. All factual claims, citations, dataset sizes, and benchmark design choices originate from the authors and were verified manually.

**Human verification:** All authors read every paragraph of the final draft; the corresponding author is responsible for the content and for any errors that remain. No reviews, citations, or empirical results were generated by AI without human verification.