# OpenReview forum: "AssamBench: Toward a Comprehensive Benchmark for Evaluating Large Language Models on Assamese"
_NortheastGenAI/2026/Workshop — NortheastGenAI 2026 Workshop Submission_

### Official Review · ~Badal_Nyalang1 · 2026-05-23
**Grounded, honest, actionable — Accept, Best Exploratory Paper candidate**

**Rating:** 8
**Confidence:** 5

**Review:**

**Relevance: Strong**
Clean T1/T3 fit. Assamese is NE India's most documented language and the benchmark gap identified is real and well-argued. The workshop CFP explicitly welcomes pilot studies and early-stage proposals.

**Plausibility: Strong**
This is the most credible proposal in the batch. The task selection is justified, annotation protocol is concrete (sources named, IAA targets set, adjudication process described), and the limitations section is unusually honest — dialect coverage gaps, annotator availability, and closed-model drift are all named directly. The pilot is small but the authors don't overclaim it.

**Novelty: Good**
The benchmark gap is real. IndicGLUE's Assamese coverage is genuinely thin, and no dedicated multi-task Assamese evaluation suite exists publicly. The error taxonomy is the strongest original contribution — particularly the ৰ vs. র script confusion angle, which is a real and underappreciated problem.

**Clarity: Strong**
Well written, well structured. The open questions in §5 are a nice touch and genuinely invite community engagement. AI disclosure is clean and appropriately scoped.

**Verdict: Accept**
Strong candidate for Best Exploratory Paper. Exactly the kind of submission this workshop should showcase. Grounded, honest, actionable, and directly useful to the NE NLP community.

*This review was generated with AI assistance and checked by the workshop chairs.*

---

### Decision · Program_Chairs · 2026-05-23

**Decision:**

Accept

**Comment:**

A strong research-oriented submission in the batch. The benchmark gap is real, the annotation protocol is concrete, and the limitations section is unusually honest about scope. The Assamese error taxonomy, particularly the ৰ vs. র script confusion finding, is a genuinely useful contribution to the NE NLP community.

Decision: Accept